# Exploring the Capability of Surface-Observed Spectral Irradiance for Remote Sensing of Precipitable Water Vapor Amount under All-Sky Conditions

5 Pradeep Khatri<sup>1</sup>, Tamio Takamura<sup>2</sup>, and Hitoshi Irie<sup>2</sup>

10 Correspondence to: Pradeep Khatri (pradeep@soka.ac.jp)

Abstract. Precipitable water vapor (PWV) is a key component of Earth's climate and hydrological systems, yet its accurate and continuous observation under varying sky conditions remains challenging. This study demonstrates the strong potential of surface-based spectral irradiance measurements for PWV retrieval across a range of atmospheric conditions using deep neural network (DNN) models trained on water vapor absorption bands. Global, direct, and diffuse spectral irradiances observed at water vapor absorption bands of 929.0–997.3 nm, 800.9–840.5 nm, and 708.1–744.6 nm by a spectroradiometer (MS-700; EKO Instruments Co., Ltd., Japan) equipped with a rotating shadow-band system were used as test data, while PWV observed by a microwave radiometer (MP-1500; Radiometrics Corporation, USA) served as reference data for model training and validation. Models incorporating global, direct, and diffuse irradiances achieved the highest accuracy, exhibiting minimal errors and closely capturing seasonal PWV variations. Notably, even models using only global irradiance—an easier and more accessible measurement—maintained high predictive performance, with low errors and robust seasonal tracking. In contrast, models trained solely on clear-sky direct irradiance with limited data showed relatively higher errors and weaker generalization, underscoring the importance of data volume and diversity in DNN models. These results highlight the effectiveness of spectral irradiance-based approaches for continuous PWV estimation across a range of atmospheric conditions. Future research should incorporate additional spectral bands sensitive to constituents like aerosols and ozone to expand retrieval capability.

#### 1 Introduction

Atmospheric water vapor plays an essential and multifaceted role in the Earth's climate system. It is not only a key driver of weather phenomena such as cloud formation and precipitation, but also a central player in atmospheric thermodynamics and energy transport. As a greenhouse gas, it absorbs and emits infrared radiation, significantly enhancing the Earth's natural greenhouse effect. Although it is easily phase-changeable gas compared to carbon dioxide or methane, its abundance and strong infrared absorption characteristics make it the most influential greenhouse gas in terms of its contribution to natural greenhouse effect. Water vapor contributes between 41% and 67% of the total natural greenhouse effect, far surpassing the contributions of other gases (Kiehl and Trenberth, 1997; Schmidt et al., 2010). However, unlike carbon dioxide and methane, water vapor is primarily a feedback rather than a direct radiative forcing, as its concentration is controlled by temperature: as global temperatures rise due to anthropogenic greenhouse gases, more water evaporates, increasing atmospheric humidity and enhancing warming further (Dessler, 2013).

The spatial and temporal distribution of water vapor varies dramatically due to its sensitivity to surface temperature, atmospheric circulation, and the phase changes of water. This variability affects not only the distribution of latent heat and cloud formation, but also radiative balance, atmospheric stability, and severe weather development. For example, increased atmospheric moisture contributes to more intense precipitation events, a trend that has been observed and is projected to

<sup>&</sup>lt;sup>1</sup>Department of Science and Engineering for Sustainable Innovation, Faculty of Science, Soka University, Hachioji-shi, Tokyo, Japan

<sup>&</sup>lt;sup>2</sup> Center for Environmental Remote Sensing, Chiba University, Chiba, Japan

continue under global warming (Trenberth et al. 2003). These characteristics make precipitable water vapor (PWV)—the total column amount of water vapor in the atmosphere above a given location—an important diagnostic variable for weather forecasting, hydrological modelling, and climate research.

Accurate observation of PWV is essential for initializing and validating numerical weather prediction (NWP) models and global climate models (GCMs). High-resolution PWV data can improve short-term precipitation forecasts, especially during convective weather events, by better representing moisture availability and transport in the lower and mid-troposphere (Van Baelen et al., 2011; Li et al., 2020; Muller et al., 2009). Furthermore, trends in PWV are used as indicators of climate change and can provide insights into shifts in the hydrological cycle, including the intensification of droughts and extreme rainfall events Allan et al., 2014). Thus, continuous and accurate monitoring of water vapor is fundamental not only for scientific understanding of the atmosphere, but also for practical applications such as agriculture, disaster preparedness, and water resource management.

Commonly used observational techniques for atmospheric water vapor—such as radiosondes, microwave radiometers, and satellite instruments—each offer distinct strengths and limitations regarding spatial and temporal resolution, vertical coverage, and observational accuracy. Radiosondes, which are balloon-borne instruments widely regarded as the standard for in situ atmospheric profiling, provide high-vertical-resolution measurements of temperature, pressure, and humidity from the surface to the stratosphere. However, they are typically launched only twice daily at synoptic hours (0 and 12 UTC) and have limited spatial coverage, particularly over oceans and in developing regions (Dirksen et al., 2014; Seidel et al., 2009). Ground-based microwave radiometers (MWRs) enable continuous, high-temporal-resolution monitoring of atmospheric temperature and humidity profiles under most weather conditions. These instruments are valuable for capturing short-term variability in atmospheric moisture, especially within the planetary boundary layer; however, their relatively high cost and calibration requirements restrict widespread deployment (Löhnert and Maier, 2012). Satellite-based passive remote sensing instruments, such as the Moderate Resolution Imaging Spectroradiometer (MODIS), the Atmospheric Infrared Sounder (AIRS), and the Infrared Atmospheric Sounding Interferometer (IASI), offer extensive global coverage and long-term observational consistency, which are essential for large-scale climate studies and operational weather forecasting. Nonetheless, they suffer from coarse temporal resolution due to their orbit characteristics and are prone to retrieval errors under cloudy and precipitating conditions, especially when using infrared channels (Schröder et al., 2019). More recently, active sensing techniques such as Global Navigation Satellite System Radio Occultation (GNSS-RO) have become increasingly important, offering highvertical-resolution humidity profiles with minimal sensitivity to cloud cover. Despite these advantages, the spatial and temporal sampling of GNSS-RO remains relatively sparse compared to passive satellite observations (Anthes et al., 2008). Collectively, these observational platforms form the foundation of current atmospheric water vapor monitoring capabilities; however, their limitations—particularly in cloudy-sky conditions—underscore the need for improved, robust, and continuous retrieval techniques that can function across all sky conditions.

Recently, sun photometers and multi-wavelength radiometers have also become crucial instruments for remotely sensing atmospheric water vapor from surface due to their ability to measure the intensity of direct solar radiation across different wavelengths. Atmospheric water vapor absorbs and scatters solar radiation at specific wavelengths, which leads to detectable variations in the amount of radiation reaching the surface, making these instruments highly effective for PWV retrieval (Campanelli et al., 2014). For example, instruments such as sun photometers or sky radiometers, which measure direct and diffuse radiation at different wavelengths, are operated in established networks such as AERONET (Holben et al., 1998) and SKYNET (Nakajima et al., 2020). These instruments are capable of inferring PWV by analysing transmittance at wavelengths, particularly around 940 nm, where water vapor has prominent absorption features (Campanelli et al., 2014). While these instruments primarily use direct radiation, retrievals are generally only possible during clear sky conditions. Moreover, the accurate retrieval of PWV from these instruments depends on effective calibration techniques as well as accurate quantification of aerosol optical thickness (AOT) at water vapor-absorbing wavelengths. AOT is typically measured at non-absorbing wavelengths and then interpolated to estimate the AOT at the 940 nm wavelength (Pérez-Ramírez et al., 2014). Similar retrieval procedures have been applied to infer PWV values using direct irradiances observed by multi-wavelength radiometers, such as Multi-Filter Rotating Shadowband Radiometers (MFRSR) (Alexandrov et al., 2009) and grating spectroradiometers (Qiao et al., 2023) as well.

Extending the remote sensing capabilities of ground-based instruments to retrieve atmospheric water vapor under all-sky conditions (both clear and cloud sky conditions) represents a significant advancement in atmospheric monitoring. Global irradiance, comprising both direct and diffuse components, can still be reliably measured using inexpensive, robust sensors.

These spectral irradiances in and within water vapor absorption bands exhibit sensitivity to changes in columnar water vapor and other atmospheric constituents, making them highly effective for retrieving PWV across a wide range of atmospheric conditions. Furthermore, the low cost, ease of deployment, and high temporal resolution of these instruments make them especially suitable for widespread use in both operational networks as well as research-grade observational campaigns. Despite such potential of using spectral irradiance measurements to retrieve water vapor under all-sky conditions, comprehensive methodologies for this task remain underdeveloped. This limitation is primarily due to the non-linear and ill-posed nature of the retrieval problem, the sensitivity to cloud properties, and the computational burden associated with traditional inversion techniques. As a promising alternative, machine learning (ML) approaches offer data-driven solutions capable of capturing complex, non-linear relationships between spectral irradiance and atmospheric water vapor, without the need for explicit physical modelling or iterative inversion (Zheng et al., 2021). Supervised ML models, trained on either simulated or observed datasets, can provide efficient, real-time retrievals that generalize well across diverse atmospheric conditions, including cloud-covered skies. These models could greatly enhance the retrieval of water vapor, especially in regions where high-cost instrumentation is scarce.

This study aims to address this research gap by developing and evaluating a ML-based framework for retrieving PWV from multi-wavelength spectral irradiance data collected under all-sky conditions. This approach will not only enhance the operational capability of existing ground-based networks such as SKYNET but also contribute to long-term climate monitoring, hydrological modelling, and satellite validation efforts in regions lacking high-cost instrumentation.

# 2 Data

This study uses data collected at Chiba (35.625°N, 140.104°E), Japan from 2012 to 2018, except for 2015 due to data unavailability, of two different types, as described below.

115

100

110

120

125

Figure 1. Observation sequence of the MS-700 spectroradiometer equipped with a rotating shadow-band, measuring total horizontal global irradiance (Irr<sub>1</sub>), partially shaded irradiances (Irr<sub>2</sub> and Irr<sub>4</sub>; 8.6° on either side of the Sun), and fully blocking the Sun's direct beam (Irr<sub>3</sub>) during each observation cycle.

# 2.1 Spectral irradiances

This study uses global, direct, and diffuse spectral irradiances observed by a spectroradiometer (MS-700, manufactured by EKO Instruments Co. Ltd., Japan) equipped with a rotating shadow-band system. The spectroradiometer measures spectral global irradiances by using a rotating shadow-band at predefined slant angles along the rotation axis (Khatri et al., 2012;

150

Takamura and Khatri, 2021). The observation sequence of the instrument is illustrated in Figure 1. The basic concept is similar to that of the Multi-Filter Rotating Shadow-band Radiometer (MFRSR) (Harrison et al., 1994); however, unlike the MFRSR, the MS-700 is a spectroradiometer capable of measuring spectral irradiance over a continuous wavelength range from 300 to 1050 nm with wavelength interval of 3.3 nm. Detailed information about the observation system is provided in Takamura and Khatri (2021). Briefly, the shadow-band-equipped spectroradiometer performs measurements at four different shadow-band positions during each observation cycle, as shown in Figure 1. Initially, the shadow-band is positioned below the horizontal plane of the observation sensor to measure total spectral irradiance (Irr<sub>1</sub>). It then rotates to a position 8.6° behind the center of the sun to measure spectral irradiance with partial shading (Irr<sub>2</sub>). Next, the shadow-band moves to fully block the sun's direct beam at the center position, measuring spectral irradiance with complete solar obstruction (Irr<sub>3</sub>). Finally, the shadow-band rotates to a position 8.6° ahead of the sun to acquire another partially shaded spectral irradiance measurement (Irr<sub>4</sub>). Through 140 this sequence, spectral irradiances corresponding to four distinct shadow-band positions are obtained in a single scan.

Figure 2. Observational data example from MS-700 for May 1, 2018 (12:00 JST), with vertical lines marking the central wavelengths of water vapor absorption.

Figure 2 presents an example of Irr<sub>1</sub>, Irr<sub>2</sub>, Irr<sub>3</sub>, and Irr<sub>4</sub> measurements collected over Chiba, Japan, at 12:00 JST on May 1, 2018, along with indications of the central wavelengths of major water vapor absorption bands. It is important to note that Irr<sub>1</sub>, Irr<sub>2</sub>, Irr<sub>3</sub>, and Irr<sub>4</sub> are measurements obtained using a horizontally mounted sensor. Consequently, these observed irradiance values are subject to cosine errors, which refer to the deviation of a radiometric sensor's angular response from the ideal cosine law (Lambert's cosine law). This law describes how irradiance should vary as a function of the angle of incoming light relative to the sensor's normal. Cosine error correction factors—typically derived from laboratory-based angular response measurements or field calibrations using solar tracking systems—are usually applied to correct these deviations and improve the accuracy of irradiance measurements, particularly at large solar zenith angles. Such corrections are crucial when using physics-based retrieval methods, which generally assume idealized input conditions. However, in supervised ML-based approaches, explicit correction for cosine errors is not necessarily required, as the ML model can implicitly learn and adjust for these systematic deviations during the training process. To simplify the retrieval procedure and minimize computational burden, this study directly utilizes the observation signals without applying cosine error corrections to the data. The cosineaffected global, diffuse, and direct irradiances used in this study can be expressed as follows (Takamura and Khatri, 2021):

$$spDHI' = Irr_{1}$$

$$spDHI' = Irr_{3} - \left(Irr_{1} - \frac{Irr_{2} + Irr_{4}}{2}\right)$$

$$spDNI' = spGHI' - spDHI'$$
(2)

$$snDNI' = snGHI' - snDHI'$$
 (3)

where spGHI', spDHI' and spDNI' represent global horizontal irradiance, global diffuse irradiance and direct normal irradiance, respectively, without applying any correction factors to observation data. 165

# 2.2 Precipitable water vapor content (PWV)

The PWV data used in this study were obtained from observations made by a microwave radiometer (Model:

MP-1500), developed by Radiometrics Corporation, USA. This passive radiometer operates in the K-band frequency range (22–30 GHz). It measures natural thermal emissions from the atmosphere at selected narrowband frequencies, primarily cantered around the 22.235 GHz water vapor absorption line. This instrument is capable of retrieving high-resolution vertical profile of water vapor and PWV under all-weather conditions. Its ability to perform accurate PWV measurement makes it highly suitable for both operational and research applications. The MP-1500 has been widely used in meteorological and climate studies, particularly for evaluating boundary layer processes, validating satellite-based retrievals, and supporting numerical weather prediction models.

# 3 Methodology

ML approaches have become increasingly important in atmospheric remote sensing for inferring geophysical quantities from complex, nonlinear input data. These data-driven models learn statistical relationships from observational datasets without relying explicitly on physical parameterizations, making them well-suited for problems where traditional inversion is challenging (Zheng et al., 2021). Several major ML techniques have been widely applied in environmental and remote sensing research. Linear regression models provide interpretable baselines but are limited by their inability to model nonlinear dependencies. Support vector machines (SVMs) with kernel functions can capture more complex relationships, but scale poorly with large datasets and are sensitive to kernel selection (Mountrakis et al., 2011). Decision tree–based ensemble methods, including random forests and gradient boosting machines (GBMs), are popular for their robustness, feature importance estimation, and relatively strong performance on tabular datasets (Belgiu and Drăgut, 2016). However, these ensemble methods can have limitations in their ability to capture complex spectral correlations and nonlinearities, which can be critical when working with high-dimensional atmospheric data with multiple features, such as spectral irradiance measurements at different wavelengths.

Deep learning approaches, particularly deep neural networks (DNNs), provide a powerful alternative with significant advantages over the aforementioned ML methods in the context of this study. Spectral irradiance data are high-dimensional and exhibit complex interactions across wavelengths due to absorption, scattering, and cloud modulation. Unlike ensemble methods, which treat each decision independently, DNNs are capable of learning hierarchical representations of data through multiple layers of neurons, allowing them to capture intricate nonlinear relationships between input features and output targets (Lecun et al., 2015). Hence, DNNs are particularly suited for applications like atmospheric remote sensing, where capturing the full complexity of the input data is crucial.

The DNN employed in this study is a fully connected multilayer perceptron (MLP) network, designed to map input spectral irradiance data to the target PWV. The input data consist of measurements of spectral irradiance across multiple wavelengths, day number of year, and solar zenith angle calculated from local time and latitude and longitude of observation site. The input data were first pre-processed and normalized before being passed through the network. The model architecture consists of several hidden layers, where each layer applies a linear transformation followed by a nonlinear activation function. At each layer, the output is calculated as

$$\mathbf{a}^{(1)} = f(\mathbf{w}^{(1)}\mathbf{a}^{(1-1)} + \mathbf{b}^{(1)}) \tag{4}$$

where,  $\mathbf{w}^{(l)}$  and  $\mathbf{b}^{(l)}$  represent the weights and biases for the l-th layer, while  $\mathbf{a}^{(l)}$  is the output activation and  $\mathbf{f}(.)$  the activation function. We used Rectified Linear Unit (ReLU) (Krizhevsky et al., 2012) activation function. This process continues through multiple layers, with each layer learning increasingly abstract features of the input data. The final output layer generates the predicted PWV value.

We used data from 2012 to 2015 of all sky conditions as model inputs. These data were divided into three subsets: training, validation, and test sets. Initially, 60% of the data was allocated to the training set, which was used to fit the

model. The remaining 40% was temporarily set aside and subsequently split evenly into validation and test sets, each comprising 20% of the total data. The validation set was used during training to monitor model performance and tune hyperparameters, while the test set was held out entirely from the training process and used only for final performance evaluation.

The model was trained using the mean squared error (MSE) loss function as

$$\mathcal{L}(\theta) = \frac{1}{N} \sum_{i=0}^{N} (\hat{y}_i - y_i)^2$$
 (5)

the model training and evaluation processes.

where,  $\hat{v}_i$  and  $v_i$  represent the predicated and true PWV values for the i-th sample, respectively,  $\mathcal{L}$  the lost function,  $\theta$ , the 220 model parameters (weights and biases), and N the number of training samples. The network parameters (weights and biases) were updated using the Adam optimizer (Kingma and Ba, 2014).

generalization and mitigate overfitting, the training process incorporated three callback mechanisms: EarlyStopping, which monitored the validation loss (val loss) and halted training if it failed to improve for 10 consecutive epochs (where an epoch represents one complete pass through the entire training dataset) while restoring the bestperforming model weights; ModelCheckpoint, which saved the model whenever a new minimum validation loss was observed to ensure retention of the optimal model; and ReduceLROnPlateau, which reduced the learning rate by a factor of 0.3 if the validation loss plateaued for 10 epochs, facilitating better convergence during later training stages. The model was trained for a maximum of 300 epochs using mini-batch gradient descent, with mini-batches supplied by train generator, a custom data 230 generator that efficiently yields batches of training samples. Simultaneously, validation data were provided via val generator to enable real-time monitoring of model performance during training.

The trained DNN model was subsequently applied to unseen spectral irradiance data from 2017 and 2018 to estimate PWV values. These estimates were then compared with ground-truth PWV measurements obtained from a microwave radiometer to evaluate the model's predictive performance. For this analysis, only data corresponding to solar zenith angles (SZA) less than 75° were used. Additionally, as part of the input data quality control, any negative values, if present, were excluded from both

It is worth mentioning that the training data for DNN model, derived from microwave radiometer observations, are subject to several well-characterized sources of error—such as solar interference, site-specific retrieval coefficient mismatches, contamination from water on the radiometer window, calibration issues, and radio frequency interference. Although the radiometer system incorporates mitigation mechanisms (e.g., temporal averaging, dew sensors, quality control flags), these error sources may still introduce both random and systematic uncertainties into the data. As the primary objective of this study is to explore the capability of surface-observed spectral irradiances for remote sensing of atmospheric water vapor under diverse sky conditions, the effects of these observational uncertainties on the trained models are not explicitly quantified.

# 4 Results and Discussion

As shown in Figure 2, three major water vapor absorption bands are present within the visible to near-infrared spectral range. To evaluate their relevance for PWV retrieval under all-sky conditions (both cloudy and clear), we developed separate predictive models for each absorption band, as well as a combined model incorporating all bands. Furthermore, given that global spectral irradiance measurements are more widely available—particularly when shadow-band systems are not employed—we assessed the model's practicality and robustness using only global spectral irradiance data. In addition, we tested the model using direct spectral irradiance data under clear-sky conditions to establish a benchmark for comparison with the all-sky and global-only retrievals.

Table 1 summarizes the wavelengths used in this study for each absorption band.

275

280

Table 1. Water Vapor Absorption Bands Used in This Study

| Band Name                       | Wavelength Range<br>(Total Number of Wavelengths) |  |
|---------------------------------|---------------------------------------------------|--|
| 940 nm centered absorption band | 929.0nm ~ 997.3 nm (22)                           |  |
| 820 nm centered absorption band | 800.9 nm ~ 840.5 nm (13)                          |  |
| 720 nm centered absorption band | 708.1 nm ~ 744.6 nm (12)                          |  |

# 4.1 Modelling using spectral global, direct, and diffuse irradiances together

Our analyses demonstrated that incorporating a certain degree of feature engineering into the input data significantly improves model performance. Accordingly, we prepared the input features as follows: for each wavelength, we computed the ratio of direct to diffuse irradiance and used it as an additional input feature. This ratio was used in combination with the spectral values of global, direct, and diffuse irradiances, all of which were scaled by the cosine of the solar zenith angle to account for solar geometry. Finally, during model training, we applied a natural logarithmic transformation to both the target variable (PWV) and all irradiance-based input features to enhance learning stability and model accuracy.

Our analyses further revealed that increasing the number of intermediate hidden layers may generally raise model complexity, thereby heightening the risk of overfitting and potentially leading to training instability or failure. Similarly, increasing the number of nodes per layer may further add model complexity and the risk of overfitting without necessarily enhancing predictive performance. From such analyses, we found that a DNN model with three hidden layers (with 32, 16, and 8 nodes) and a single-node output layer performs best on our datasets. Thus, we used model of this configuration in this study.

#### 4.1.1 Loss function evaluation

Figure 3. Training and validation losses for spectral direct, diffuse and global irradiance-based input features for band centered at (a) 940 nm, (b) 820 nm, (c) 720 nm, and (d) all of them. The arrows indicate the minimum validation loss.

Figure 3 presents an evaluation of model performance through training and validation loss curves across three spectral bands centered at 940 nm, 820 nm, and 720 nm, and combination of all of them under all-sky conditions (including both clear and cloudy skies). The variation in the number of training epochs among subplots reflects the use of an EarlyStopping criterion, as detailed in the methodology. Each subplot adopts a dual-panel layout: the upper panel displays a broader loss range (0.06–0.09), while the lower panel highlights a finer range (0.002–0.02), allowing for a detailed inspection of convergence behavior. In each subplot, the red line represents the training loss, and the blue line shows the validation loss, both calculated using the MSE loss function using an equation mentioned above. Arrows indicate the minimum validation loss during the training process, representing the point of optimal generalization performance.

Across all three bands and their combination, the loss curves exhibit a rapid initial decline within the first 10–15 epochs, indicating efficient learning during early training. This is followed by a gradual flattening of the curves, suggesting convergence is generally achieved within 40–70 epochs depending on the spectral band. The relatively small difference between training and validation losses throughout the training period suggests that the models maintain strong generalization capacity and are not significantly overfitting the training data. Minor fluctuations in the validation loss are normal and are due to the stochastic nature of mini-batch training and variability in the validation data.

For the 940 nm band (Figure 3a), the model achieves the lowest overall training and validation losses among the three selected bands. This outcome is consistent with the well-known fact that the 940 nm band is the strongest near-infrared water vapor absorption feature (Gueymard, 1995), providing high sensitivity to columnar water vapor content even under cloudy conditions. The model's strong performance across mixed-sky inputs highlights the robustness of this spectral signal and demonstrates the DNN's ability to capture the underlying physical relationship effectively. For the 820 nm band (Figure 3b), the model also shows good convergence, although the final loss values are slightly higher compared to the 940 nm band. While the 820 nm band exhibits moderate water vapor absorption strength(Gueymard, 1995), it still carries valuable PWV information. Data of this absorption band is particularly useful in scenarios where the 940 nm band is partially obstructed or unavailable. The consistent alignment between training and validation curves suggests that the model trained on this band can still generalize well under diverse atmospheric conditions. The 720 nm band (Figure 3c), which represents the weakest of the three absorption features analyzed (Gueymard, 1995), also leads to a reasonably converged model. However, it shows relatively higher and slightly more variable validation losses. This is expected due to the weaker absorption of water vapor in the visible range and the stronger influence of confounding factors such as aerosols, clouds, and surface reflectance (Khatri et al., 2016, 2019; Nakajima et al., 2020). Nevertheless, the narrow gap between training and validation losses indicates that the model remains well-regularized and capable of extracting meaningful information, even for such weak absorption band. Crucially, when all three spectral bands were combined as a unified input, the DNN model achieved exceptionally low and stable losses, comparable to or potentially surpassing the performance of the best individual band (940 nm). This outcome highlights a powerful synergistic effect, where the integration of diverse spectral information leads to enhanced accuracy and robustness. The slightly longer convergence time for the combined band suggests the model requires more iterations to fully exploit the increased complexity and interdependencies within the richer input feature space.

#### 4.1.2 Validation through comparison between predicted and unseen ground truth values

#### 4.1.2.1 Scatter plot validation

315

325

Figure 4. Comparison between predicted and observed PWV values for spectral bands centered at (a) 940 nm, (b) 820 nm, (c) 720 nm, and (d) all of them using DNN models trained on all-sky spectral irradiance data of global, direct, and diffuse. The predictions were made using models that achieved the minimum validation losses (as shown in Figure 3). The red dashed line indicates the 1:1 line.

Figure 4 shows the relationship between the predicted and ground-truth PWV values for the three spectral bands centered at (a) 940 nm, (b) 820 nm, (c) 720 nm, and for (d) combination of all of them. These predictions were made using models that

355

achieved the lowest validation losses as indicated by the arrows in Figure 3. Each panel presents a 2D density scatter plot comparing predicted PWV (y-axis) values against PWV (x-axis) values observed by microwave radiometer, with the red dashed line denoting the 1:1 reference line (ideal prediction).

The 940 nm centered band (Figure 4a) yields the highest performance among three different individual spectral configurations with a Root Mean Square Error (RMSE) of 0.174 cm and correlation coefficient (R2) of 0.978. Visually, Figure 4a 330 demonstrates a very tight clustering of data points closely aligned with the 1:1 reference line, indicating minimal scatter and high predictive fidelity across the entire range of actual PWV values. This strong performance is consistent with its corresponding loss curves in Figure 3a, where the model exhibited the lowest training and validation losses and stable convergence among three different individual spectral configurations. This relatively high accuracy is attributed to the strong water vapor absorption feature at 940 nm, which provides high spectral sensitivity to column water vapor under a wide range 335 of atmospheric conditions. The low RMSE signifies that the model's predictions are consistently very close to the true values, while the near-perfect R<sup>2</sup> indicates that almost all the variability in observed PWV is explained by the model's output. For the 820 nm band (Figure 4b), the model also performs commendably, achieving RMSE of 0.237 cm and R<sup>2</sup> of 0.966. While slightly less accurate than the 940 nm band, this result aligns with Figure 3b, which showed effective model convergence and

340 low loss values, albeit marginally higher than the 940 nm case. Figure 4b shows a good alignment with the 1:1 line, though with a slightly wider spread of points compared to the 940 nm band. Although the 820 nm band has weaker absorption, it is still useful for retrieving PWV, especially when 940 nm data are unavailable or less reliable due to instrument issues or very strong absorption that could potentially saturate the signal. Its performance confirms its utility as a valuable alternative or complementary data source, offering a reliable measurement even with a moderately weaker signal.

345 The 720 nm band (Figure 4c) achieves a comparable RMSE of 0.237 cm and R<sup>2</sup> of 0.967, similar to the 820 nm result. This reflects the model's robustness in learning from the weaker absorption band, although its performance is slightly more variable, as also observed in the corresponding loss curve (Figure 3c), which showed higher fluctuation in validation loss. Because this band is less sensitive to water vapor, the predictions are more scattered around the 1:1 line—especially at higher PWV levels likely due to weaker signals and more noise from other factors, such as aerosols, clouds, surface etc. While the DNN 350 demonstrates a remarkable ability to extract meaningful information even from such a challenging signal, the trade-off between signal strength and prediction precision becomes evident, leading to greater uncertainty at higher PWV values where the water vapor signal is less distinct relative to background noise.

The combined spectral band approach (Figure 4d) visually demonstrated superior performance, appearing as good as or even slightly better than the 940 nm band alone. This is reflected with RMSE of 0.157 cm and an R<sup>2</sup> of 0.982. Figure 4d displays a remarkably tight clustering of predicted values around the 1:1 line, indicating exceptional accuracy and linearity across the entire PWV range. This highlights the significant advantage of multi-spectral data fusion. By integrating information from bands with varying sensitivities, the DNN can leverage complementary data. For instance, while the 940 nm band provides the primary, strong water vapor signal, the 720 nm band, despite its weak water vapor absorption, is more sensitive to confounding factors like aerosols and clouds. The DNN can implicitly use this contextual information to correct for their influence on the 360 stronger absorption bands, effectively performing an "internal atmospheric correction". This leads to a more robust and accurate PWV retrieval, particularly under complex all-sky conditions where multiple atmospheric components are at play. This synergistic effect means the combined input provides a richer, more comprehensive representation of the atmospheric state, allowing the DNN to achieve optimal accuracy and enhanced resilience against atmospheric variability.

#### 4.1.2.2 Temporal validation across varying weather and sky conditions

Figure 5 provides a detailed temporal validation of the DNN models' performance by comparing predicted and actual PWV values on a monthly basis. Presented as box plots, this figure offers a comprehensive view of the central tendency (median), spread (interquartile range), and variability of both actual (blue) and predicted (red) PWV values for each month of the year. This monthly comparison serves as a crucial complement to the overall accuracy metrics (RMSE, R2) presented in Figure 4 and the training dynamics observed in Figure 3, revealing how consistently the models perform across seasonal atmospheric

A general observation across all subplots in Figure 5 is the clear seasonal cycle of PWV, with higher values typically observed during the summer months (e.g., July-September) and lower values during the winter months (e.g., January-March). This expected pattern is well captured by the actual PWV data, and the key evaluation lies in how accurately the predicted values track these monthly variations.

405

415

For the 940 nm centered band (Figure 5a), the box plots demonstrate remarkably close agreement between the actual and predicted PWV values across all months. The medians of the predicted (red) boxes consistently align well with those of the actual (blue) boxes, and their interquartile ranges largely overlap. Even the whiskers, representing the full range of data, show strong correspondence.

The 820 nm centered band (Figure 5b) also shows good overall agreement between predicted and actual monthly PWV. The predicted box plots generally follow the seasonal trend of the actual data. However, upon closer inspection, especially during months with higher PWV or greater atmospheric variability, subtle deviations or slightly larger discrepancies between the predicted and actual distributions might be observed compared to the 940 nm band. This visual observation is consistent with the slightly higher RMSE (0.237 cm) and lower R<sup>2</sup> (0.966) for the 820 nm band presented in Figure 4b, and the marginally higher loss values in Figure 3b.

In the case of the 720 nm centered band (Figure 5c), the monthly comparison reveals more noticeable discrepancies, particularly in months with higher PWV levels or more dynamic atmospheric conditions. While the general seasonal trend is captured, the predicted box plots might show less perfect alignment with the actual data, potentially exhibiting larger differences in medians, wider spreads, or less accurate capture of extreme values. This increased scatter and variability in monthly predictions align directly with the larger scatter observed in Figure 4c's density plot and the more variable validation losses shown in Figure 3c. These discrepancies underscore the inherent challenges of retrieving PWV from a very weak absorption band, where confounding factors like aerosols, clouds, and surface reflectance exert a stronger influence, leading to less consistent performance across diverse monthly atmospheric states.

Finally, the combination of all bands (Figure 5d) visually presents the most compelling results in terms of monthly tracking of PWV. The alignment between the predicted and actual box plots is remarkably tight, often appearing as good as, if not superior to, the 940 nm band alone. The medians are highly congruent, and the interquartile ranges and overall spread of the predicted values closely mirror the actual data across all months. This outstanding consistency in capturing monthly variations strongly corroborates the best overall RMSE (0.157 cm) and R<sup>2</sup> (0.982) metrics reported for the combined approach in Figure 4d, as well as the very low and stable loss curves in Figure 3d. This outcome powerfully demonstrates the synergistic advantage of multi-spectral data fusion.

Overall, Figure 5 provides essential temporal validation across varying weather and sky conditions, confirming that the performance characteristics observed in the overall accuracy metrics (Figure 4) and training dynamics (Figure 3) translate consistently across monthly variations.

Figure 5. Monthly comparison of DNN-predicted versus actual PWV values for spectral bands centered at (a) 940 nm, (b) 820 nm, (c) 720 nm, and (d) combination of all of them.

# 410 4.2 Modelling using only spectral global irradiance or spectral direct irradiance

When modelling using either only spectral global irradiance or only spectral direct irradiance as the primary radiation component, we largely followed the same feature engineering approach described in Section 4.1. Specifically, for the global-only model, features corresponding to spectral direct and diffuse irradiance were excluded; conversely, for the direct-only model, features related to spectral global and diffuse irradiance were omitted. Additionally, since only a single radiation component was used in each case, the ratio of spectral direct to diffuse irradiance was also excluded from the feature set.

For modelling with spectral direct irradiance as the primary input, we used data corresponding exclusively to clear-sky conditions. These conditions were identified based on clear-sky periods detected by a co-located sky radiometer from the SKYNET network (Hashimoto et al., 2012; Khatri et al., 2014, 2019; Nakajima et al., 2020) installed at the same site of shadow-band spectroradiometer. Cloud screening for the sky radiometer data was performed using the algorithm of Khatri and Takamura (2009). To further reduce the possibility of data affected by very thin clouds, we applied an additional filter to exclude samples with aerosol optical thickness (AOT) at 500 nm greater than 0.5. While this additional screening helped to more strictly ensure clear-sky conditions, it also significantly reduced the number of available data samples.

Table 2 summarizes the number of data samples used for model training, model validation, and evaluation with unseen data when using only spectral global irradiance or only spectral direct irradiance as the primary radiation component.

Table 2. Summary of Data Counts Used for Modelling and Validation

|                      | - V      |            |        | 8                            |
|----------------------|----------|------------|--------|------------------------------|
| Sky conditions       |          | Modelling  |        | Validation Using Unseen Data |
|                      | Training | Validation | Test   |                              |
| All-sky conditions   | 57,736   | 19,245     | 19,246 | 97,274                       |
| Clear-sky conditions | 3,022    | 1,008      | 1,008  | 7,259                        |

#### 4.2.1 Loss function evaluation

Figure 6. Minimum validation loss and the corresponding training loss for models using (a) only global irradiance and (b) only clear-sky direct irradiance as input features, across wavelength bands centered at 940 nm, 820 nm, 720 nm, and their combination.

Figure 6 provides a direct visual comparison of predictive model performance when input features are restricted to either only spectral global irradiance (Figure 6a) or only spectral direct irradiance (Figure 6b). In Figure 6a, which illustrates models trained exclusively with global irradiance, a consistent pattern of low minimum validation losses and corresponding training losses is observed across all individual wavelength bands (940 nm, 820 nm, 720 nm) and with slightly higher values for their combination. The loss values generally range from approximately 0.010 to 0.017. A key characteristic is the remarkable proximity between the validation loss and its corresponding training loss for each configuration. This suggests that the models have effectively learned the fundamental patterns inherent in global irradiance data, rather than merely memorizing noise or specific examples from the training set.

Conversely, Figure 6b, depicting models using only clear-sky direct irradiance, presents a different picture. These models generally exhibit higher overall loss values, with training losses ranging approximately from 0.013 to 0.017, and validation losses slightly lower, from 0.010 to 0.013. A more noticeable, and somewhat complex, relationship between training and validation loss is apparent; for instance, at the 820 nm and 720 nm centered bands, the training loss is notably higher than the validation loss. While typical overfitting is characterized by training loss being significantly lower than validation loss, this observed scenario, where training loss is higher or not consistently lower, suggests that the model might be struggling to adequately learn the complex patterns within the training data itself, potentially indicating a degree of underfitting.

The most significant factor driving this stark difference in performance between the two different radiation data sets is the data volume. As summarized in Table 2, the global irradiance models benefited from a substantially larger dataset for "All-sky conditions". This ample data provided the models with a rich and diverse set of examples, enabling them to learn more comprehensive and robust patterns. In contrast, the clear-sky direct irradiance models were constrained by a significantly smaller dataset for "Clear-sky conditions". The reduced data volume makes it challenging for the model to learn robust, generalizable features, increasing the risk of suboptimal learning or underfitting, where the model cannot adequately fit the available data. This highlights that while the choice of specific irradiance components is relevant, the sheer quantity and quality of data for each component are paramount for achieving robust and reliable predictive models.

#### 4.2.2 Validation through comparison between predicted and unseen ground truth values

Figure 7. (a) Root Mean Square Error (RMSE) and (b) coefficient of determination (R²) for models using only global irradiance and only clear-sky direct irradiance as input features, evaluated across wavelength bands centered at 940 nm, 820 nm, 720 nm, and their combination. The predictions were generated using the models that achieved the minimum validation loss, as shown in Figure 6.

Figure 7 provides crucial validation of the predictive models' performance on unseen data, building upon the loss evaluations presented in Figure 6, by utilizing RMSE in panel (a) and R² in panel (b). In Figure 7a, models using only global irradiance from individual absorption bands consistently exhibit lower RMSE values (e.g., ~0.26 cm for 940 nm band), indicating higher predictive precision and smaller average errors on unseen data. This directly aligns with the low training and validation losses observed for global irradiance models in Figure 6a. Conversely, models using only clear-sky direct irradiance show generally higher RMSE values (e.g., ~0.32 cm for 820 nm band) at relatively stronger absorption bands, signifying larger prediction errors, which is consistent with their higher overall loss values in Figure 6b. Similarly, in Figure 7b, models with global irradiance inputs consistently achieve higher R² values (ranging from ~0.91 to ~0.95), demonstrating that a larger proportion of the variance in the data is explained by these models, indicating a better fit and stronger predictive capability. These high R² values further validate the robust generalization inferred from the low and converging losses in Figure 6a. In contrast, clear-

490

500

505

sky direct irradiance models show consistently lower R² values (ranging from ~0.90 to ~0.91), implying a weaker fit and less explanatory power, which correlates with their less ideal loss profiles in Figure 6b. This consistent pattern across both Figure 6 and Figure 7 underscores that the data volume is the primary factor driving model performance; the global irradiance models, benefiting from a substantially larger dataset (over 97,000 total samples for "All-sky conditions"), consistently outperform the clear-sky direct irradiance models, which were constrained by a significantly smaller dataset (only about 7,200 total samples for "Clear-sky conditions") due to stringent filtering. This highlights that ample data can enable models to learn more robust patterns, leading to superior accuracy and generalization on unseen data.

Figure 8. Monthly comparison of predicted and observed PWV values for the spectral band centered at 940 nm, using models based on (a) only global irradiance and (b) only clear-sky direct irradiance as input features.

Figure 8 offers a detailed monthly comparison of predicted and observed PWV values specifically for the 940 nm centered spectral band, serving as a representative illustration of the models' performance. Similar to Figure 5, panel (a) showcases results for models utilizing only global irradiance as input features, while panel (b) presents results for models based on only clear-sky direct irradiance.

In Figure 8a, which illustrates the performance of models trained with only global irradiance, a very good alignment is evident between the predicted (red boxes) and actual (blue boxes) PWV values across all months. Both the observed and predicted data clearly capture the expected seasonal cycle, with PWV values generally increasing from winter (e.g., months 11-2, typically around 1-1.5 cm) to summer peaks (e.g., months 7-8, reaching approximately 4-5 cm) before declining again. Crucially, the medians (yellow lines) of the predicted values closely track those of the actual values for each month. Furthermore, the interquartile ranges (the boxes themselves) and the overall spread indicated by the whiskers demonstrate a very good overlap and consistency. This visual evidence of high fidelity between predictions and observations directly reinforces the quantitative metrics previously discussed in Figure 6a and Figure 7. The global irradiance models corresponding to stronger absorption bands consistently exhibited lower training and validation losses (Figure 6a), coupled with lower RMSE and higher R<sup>2</sup> values (Figure 7a and 7b), collectively signify superior predictive precision and strong generalization capabilities on unseen data.

Conversely, Figure 8b, which depicts models based on only clear-sky direct irradiance, reveals relatively less precise alignment between predicted and actual PWV values. While the general seasonal trend is still discernible, noticeable discrepancies emerge, particularly during months characterized by higher PWV values (e.g., July to September). The predicted medians (red yellow lines) do not consistently align as closely with the actual medians (blue yellow lines) as observed in Figure 8a. Moreover, the spread of the predicted values (red boxes and whiskers) sometimes deviates more significantly from the observed spread,

indicating that these models struggle to capture the full variability and distribution of PWV as accurately as their global irradiance counterparts. This visual assessment of reduced precision is entirely consistent with the higher overall loss values observed for clear-sky direct irradiance models in Figure 6b, as well as their higher RMSE and lower R<sup>2</sup> values in Figure 7a and 7b.

The pronounced difference in performance between the two sets of models, as vividly demonstrated in Figure 8, is primarily attributable to the data volume available for training as summarized in Table 2 and discussed above.

# 515 **5 Conclusions**

This study highlights the strong potential of surface-observed spectral irradiances for retrieving atmospheric water vapor (PWV) across varying sky conditions, including both clear and cloudy atmospheres. By applying deep learning techniques to spectral irradiance data within three key water vapor absorption bands, we demonstrated that high-fidelity PWV retrieval is achievable even in the presence of atmospheric complexities such as clouds, aerosols, and variable solar geometry. The inclusion of global, direct, and diffuse irradiances provided enhanced sensitivity to radiative transfer characteristics, enabling the model to effectively disentangle water vapor signals from confounding influences, leading to robust performance and accurate seasonal predictions.

However, a deeper investigation into models utilizing single radiation components revealed a critical dependency on data availability. Even in observational setups where only global spectral irradiance is available—such as in systems lacking shadow bands—models retained substantial predictive power. These global irradiance-only models consistently exhibited low training and validation losses, along with superior Root Mean Square Error (RMSE) and coefficient of determination (R²) values, confirming their high predictive precision and strong generalization capabilities. Their monthly predictions of PWV closely aligned with actual observations, accurately capturing seasonal cycles. This finding is particularly valuable for expanding the operational viability of PWV monitoring in cost-constrained or logistically limited environments.

In contrast, models trained exclusively on clear-sky direct irradiance, despite the purity of their input data, generally displayed higher overall losses, larger RMSE values, and lower R² values. Their monthly PWV predictions showed less precise alignment with observations, particularly during periods of high water vapor, indicating a struggle to fully capture the data's variability and generalize effectively. This performance disparity underscores a major finding: the volume of training data is a paramount factor influencing model accuracy and generalization. The superior performance of the global irradiance models is directly attributable to the substantially larger and more diverse dataset available for all-sky conditions. This ample data enabled the models to learn comprehensive and robust patterns, mitigating overfitting and enhancing their ability to generalize to unseen scenarios. Conversely, the clear-sky direct irradiance models were constrained by a significantly smaller dataset, a consequence of stringent filtering to ensure clear-sky purity. This data scarcity limited their capacity to learn sufficient edge cases and diverse scenarios, thereby hindering their generalization and resulting in higher prediction errors.

Across all configurations, the models demonstrated good generalization and stability over seasonal cycles, supporting their applicability in diverse climatic contexts. The results collectively indicate that surface-based spectral observations, when paired with robust machine learning models, offer a scalable and weather-resilient approach to atmospheric water vapor sensing. This capability can complement satellite-based systems, enhance ground-truth validation networks, and contribute to improved understanding of hydrological and radiative processes in Earth's atmosphere. Ultimately, these findings support broader adoption of spectral irradiance-based monitoring strategies in climate science, meteorology, and remote sensing applications, while also emphasizing the critical need for sufficient and diverse training data to unlock a model's full predictive potential.

**Author Contributions:** Conceptualization, P. K., T. T, and H. I.; methodology, P. K.; software, P. K.; formal analysis, P. K.; investigation, P. K.; resources, T. T. and H. I.; writing—original draft preparation, P. K.; writing—review and editing, P. K., T.T, and H. I.; funding acquisition, P. K. All authors have read and agreed to the published version of the manuscript.

Funding: This research was funded by JSPS KAKENHI Grant Number 24K07129.

Data Availability Statement: Precipitable water vapor data used in this study are available for download from the SKYNET homepage (<a href="http://atmos3.cr.chiba-u.jp/skynet/data.html">http://atmos3.cr.chiba-u.jp/skynet/data.html</a>). The spectral irradiance data can be made available upon request to the corresponding author and co-authors.

Conflicts of Interest: Authors have no conflict of interest.

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
