# Peer review of "Exploring the Capability of Surface-Observed Spectral Irradiance for Remote Sensing of Precipitable Water Vapor Amount under All-Sky Conditions"

_EGUsphere, 2025_

## Author Comment (AC1)

**Replies to comments of Reviewer 1**

Authors would like to express sincere gratitude to an anonymous reviewer for his/her valuable comments and suggestions. Our manuscript has greatly benefited from these insights. We have carefully revised the manuscript, taking all comments into account. Our responses to the reviewer's comments are provided below.

**GENERAL COMMENTS**

The study by Khatri et al. investigates precipitable water vapour (PWV) retrieval using near-infrared spectral irradiances measured by a ground-based spectroradiometer (EKO MS-700) and deep neural network (DNN) models trained with a microwave radiometer (MWR) as the reference instrument. The manuscript explores the algorithm potential by testing individual spectral bands as well as their combination, and by evaluating spectral irradiances in different geometrical configurations (direct, global, diffuse, separately and jointly). The models generally demonstrate good performance, except for the one based only on direct irradiances under clear-sky conditions, which is mainly attributed to the limited amount of training data available.

Overall, the manuscript is clearly written and represents a valuable methodological study that could form the basis for new PWV retrieval approaches. However, as the primary motivation of the work is to extend monitoring to locations lacking reference instruments, certain concerns regarding the practical applicability of the proposed algorithm should be addressed before publication. These are detailed in the sections below.

→ We sincerely appreciate the reviewer's thoughtful and constructive comments. These have allowed us to improve the clarity of the manuscript, including the study's motivation and its practical applicability. In response, we have revised the relevant text and added additional discussion and figure where appropriate. Detailed responses to each comment are provided below.

**SPECIFIC COMMENTS**

- 1. To my understanding, the main motivation of the study is to develop an algorithm enabling the wider deployment of accurate PWV measurements using "inexpensive, robust sensors" (line 92), such as solar spectroradiometers. According to the authors, this would "enhance the retrieval of water vapour, especially in regions where high-cost instrumentation is scarce" (line 105) and be "valuable for expanding the operational viability of PWV monitoring in cost-constrained or logistically limited environments" (lines 528–529). However, the results indicate that a large amount of reference data, presumably covering a broad range of atmospheric conditions, must be collected before the DNN model can be applied, e.g. the training set used in the paper spans four years. This, in practice, requires co-located, high-cost reference instrumentation such as MWRs for a long time. Therefore, I suggest that the authors clarify several points to better demonstrate whether the proposed algorithm can be implemented under real-world conditions:
- → Thank you for this important comment. As indicated by the title and described in the abstract, this study aims to assess the feasibility of retrieving PWV under all-sky conditions using surface-observed spectral irradiances. The motivation for such study stems in part from the fact that most well-established surface-based networks—such as AERONET (Holben et al., 1998), SKYNET (Nakajima et al., 2020), A-SKY (Irie et al., 2011; Mizobuchi et al., 2025), and NDACC (De Mazière et al., 2018)—currently provide PWV retrievals only under clear-sky conditions. The statements in the "Introduction" regarding inexpensive sensors, enhanced retrieval capability in data-sparse regions, and potential applicability in cost-constrained environments were intended to

highlight the broader, long-term benefits that could arise if such all-sky retrieval capability could be established. They were not intended to imply that low-cost operational deployment is the primary motivation of the present work. Rather, the present study focuses on assessing the scientific feasibility and performance of an all-sky PWV retrieval approach derived from surface spectral irradiance measurements.

**For clarity, we have revised the original sentences in the manuscript by adding new text to better explain the study's aim, motivation, and potential long-term benefits (Lines 54-103).**

In this study, MWR observations were used as the PWV reference because they were continuously available at the site and provide reliable measurements. However, the methodological framework does not require MWRs specifically. In principle, the model can be trained using PWV data from other standard reference sources, such as radiosonde observations or GNSS-based PWV retrievals, depending on data availability at the location.

**For clarity, we have added the following sentences in the revised manuscript (Lines 258-261)**

In this study, MWR data were used because they were available at the site as a reliable PWV reference. However, the method can operate without relying specifically on MWR measurements. The model can be trained using PWV data from other standard reference sources, such as radiosonde observations or GNSS-based PWV retrievals, depending on data availability at a given location.

We also acknowledge that larger and more diverse training datasets generally improve the robustness of DNN models. The four-year dataset used here reflects the length of the time series available to us, rather than a methodological requirement. The amount of training data needed is expected to depend on local atmospheric variability and the intended operational context. For example, sites with limited seasonal variation may require comparatively shorter training periods, whereas locations with stronger variability may need longer datasets. A systematic evaluation of minimum data requirements is an important subject for future investigation.

**For clarity, we have added the following sentences in the revised manuscript (Lines 261-263)**

Although larger and more diverse training datasets generally improve model robustness, the four-year dataset used here reflects data availability rather than a strict requirement of the method. The necessary training period can be adapted to local atmospheric variability and operational needs.

- 1a. The study focuses on a single site for both training and testing. Is the DNN model site-dependent? Must the algorithm be trained under conditions similar to those expected during future measurements? From my understanding, the DNN does not explicitly distinguish between instrument-dependent and site-dependent features, and relocating the spectroradiometer to a site with substantially different conditions would likely degrade retrieval accuracy.
- → Thank you for this valuable comment. We agree that the model does not explicitly separate site-dependent and instrument-dependent features; rather, it learns statistical relationships between the

input spectral irradiances and PWV. As these relationships can be influenced by atmospheric and surface conditions, relocating the same instrument to a site with substantially different conditions without any model adaptation may indeed reduce retrieval accuracy.

However, this does not limit the applicability of the approach to a single site. The model can be extended to new locations through several strategies. The most straightforward option is to retrain the model using PWV values obtained from locally available reference sources, such as radiosondes or GNSS-based PWV retrievals. Except it, a more efficient and widely used strategy is transfer learning (Pan et al., 2010; Weiss et al., 2016), where a model trained on one dataset is adapted to a new dataset by fine-tuning selected layers. In this context, a model trained at one site can be adapted to a new site using only a comparatively small amount of local irradiance–PWV data, thereby reducing the need for long-term co-located reference observations while maintaining retrieval performance. Such transfer learning approaches have been widely applied in ML-based remote sensing retrievals (e.g., Chen et al., 2025; Dong et al., 2024; Gupta et al., 2024). These studies demonstrate that model adaptation with limited site-specific data can effectively maintain retrieval accuracy when applying trained models to new environments.

**We have summarized the above discussion in the revised manuscript as follows (Lines 264-270).**

Furthermore, the model learns statistical relationships between spectral irradiances and PWV, which can be influenced by local atmospheric and surface conditions. Relocating the instrument to a site with substantially different conditions may reduce accuracy. However, the model can be adapted to new locations through retraining with local reference PWV data or, more efficiently, via transfer learning (Pan et al., 2010; Weiss et al., 2016), in which a pre-trained model is finetuned using a relatively small amount of site-specific data. Recent studies (e.g., Chen et al., 2025; Dong et al., 2024; Gupta et al., 2024) show that such adaptation strategies effectively preserve retrieval performance when transferring ML-based remote-sensing models across sites.

- 1b. For new instruments or sites, how long would the training phase need to be? If several years of reference data are required before deployment, would such a training procedure be practical for large-scale implementation of the technique?
- → The duration of the training period for a new instrument or site depends primarily on the variability of local atmospheric and surface conditions rather than a fixed period. While longer datasets generally improve DNN robustness, sites with relatively stable conditions may require only a few months of high-quality reference data. In this study, as a first step, we developed and evaluated the procedure using data from a single site to establish feasibility and demonstrate that surface spectral irradiances can effectively capture PWV information under well-characterized conditions. Once feasibility is confirmed, the method can be extended to other sites, and the required training period can be adjusted based on local atmospheric variability. Moreover, as described above, a transfer learning strategy can further reduce the amount of site-specific reference measurements needed, supporting efficient adaptation to new instruments and/or locations.

1c. How stable is the spectroradiometer expected to be, and how frequently should the model be recalibrated (e.g. to account for instrumental drift or degradation)?

→ Although modern spectroradiometers are generally stable, small instrumental drifts or gradual degradation may occur over time. Fundamentally, our retrieval framework assumes that the input spectral irradiances are properly calibrated, and any systematic changes in instrument response should ideally be addressed at the calibration or quality-control level. At the same time, in practical applications, the model learns robust statistical relationships between spectral irradiances and PWV, and these relationships may remain largely valid even if minor changes in instrument response happen. In such situations,, instead of full retraining, periodic fine-tuning of the model using a limited amount of recent reference data—such as a few weeks or months of measurements—can typically correct for instrumental drift and maintain retrieval accuracy. This approach can preserve the core learned patterns of the DNN while adapting to slow changes in instrument performance, making the method practical for long-term, operational deployment without compromising reliability.

**We have summarized the above discussion in the revised manuscript as follows (Lines 271-276).**

Finally, although modern spectroradiometers are generally stable, minor instrumental drifts can occur over time. Our retrieval framework assumes that the input spectral irradiances are properly calibrated, and any systematic changes in instrument response should ideally be handled at the calibration or quality-control level. In practical operation, however, the DNN learns robust relationships between spectral irradiances and PWV that can remain largely stable even when small residual drifts occur. In such cases, periodic fine-tuning with a limited amount of recent reference data may be typically sufficient to adjust for these gradual shifts and maintain retrieval accuracy.

- 2. Related to points 1b–1c: the manuscript states that the day number of the year is included as an input variable in the training. This effectively provides the algorithm with a prior on the most likely atmospheric conditions for a given time of year. Could the authors assess the relative importance of all input variables in the DNN, including the day number? How can they be confident that the agreement shown in Figs. 5 and 8 is not partly driven by the climatology implicit in the training data? Moreover, what would happen if the model were applied to a site with a very different PWV climatology?
- → Thank you for this important comment. As suggested by the reviewer, we evaluated the relative importance of all input variables used in the DNN by applying the SHAP (SHapley Additive exPlanations) method (Lundberg and Lee, 2017). SHAP provides a unified, theoretically grounded framework based on cooperative game theory to quantify the contribution of each input feature to the model output by computing Shapley values, defined as the average marginal contribution of a feature across all possible feature combinations. This method has been widely used in recent ML-based atmospheric and remote-sensing studies to assess the relative influence of input variables on model predictions (e.g., Lundberg et al., 2020; Zhao et al., 2024).

In our study, the major input features include solar-zenith-angle-normalized spectral irradiances at multiple wavelengths and the day number of the year (DOY). When using spectral global, direct, and diffuse irradiances, we additionally incorporated the ratio of direct to diffuse irradiances as supplementary feature. Tables R1-R3 below summarize the SHAP values obtained for all input features under different modeling strategies (i.e., using global, direct, and diffuse irradiances jointly, or using only global or only direct irradiances), as described in the manuscript.

The SHAP values for the radiative components shown in the table represent averages across all wavelengths, given that wavelength-by-wavelength SHAP reporting would be excessively lengthy.

Table R1. Relative SHAP values (in %) of the input features for the DNN models trained using spectral direct, diffuse, and global irradiances of different absorption bands, together with the direct-to-diffuse irradiance ratio (Ratio) and the day of year (DOY)

| · ·                    |           |            |           |          |        |
|------------------------|-----------|------------|-----------|----------|--------|
| Centered band/Features | Direct(%) | Diffuse(%) | Global(%) | Ratio(%) | DOY(%) |
| 940 nm                 | 23.80     | 17.89      | 35.67     | 22.42    | 0.22   |
| 820 nm                 | 18.30     | 19.20      | 48.39     | 13.97    | 0.14   |
| 720 nm                 | 20.03     | 15.24      | 44.69     | 19.95    | 0.08   |
| Combined (all above)   | 27.18     | 19.43      | 32.92     | 20.35    | 0.11   |

Table R2. Relative SHAP values (in %) of the input features for the DNN models trained using spectral global irradiances of different absorption bands together with day of year (DOY)

| Centered band/Features | Global(%) | DOY(%) |
|------------------------|-----------|--------|
| 940 nm                 | 98.65     | 1.35   |
| 820 nm                 | 99.67     | 0.33   |
| 720 nm                 | 99.51     | 0.49   |
| Combined (all above)   | 99.54     | 0.46   |

Table R3. Relative SHAP values (in %) of the input features for the DNN models trained using spectral direct irradiances of different absorption bands together with day of year (DOY)

| Centered band/Features | Global(%) | DOY(%) |
|------------------------|-----------|--------|
| 940 nm                 | 97.84     | 2.16   |
| 820 nm                 | 99.19     | 0.81   |
| 720 nm                 | 99.15     | 0.85   |
| Combined (all above)   | 98.49     | 1.51   |

These tables show that the DNN predictions are overwhelmingly driven by the spectral irradiances themselves. In Table R1, global irradiance consistently provides the largest contribution, followed by direct and diffuse components, while the direct-to-diffuse ratio contributes moderately and DOY contributes almost negligible. Tables R2 and R3 further confirm that when only global or only direct irradiances are used, their SHAP importance exceeds 97–99%, with DOY accounting for less than 2%.

As shown by those statistical anlayses, DOY can have very low importance because it only provides indirect seasonal information, whereas the spectral irradiances directly capture the actual atmospheric state (water-vapor absorption, scattering, SZA effects, etc.). Overall, the results indicate that the spectral radiometric information dominates the model performance rather than DOY.

We have summarized the above discussion in the revised manuscript by including Table R1 as Table 2 (), and by adding the following sentences in the section discussing the results of Figure 5 (Lines 459-474).

Since DOY is included as one of the input features in our DNN models, it is important to examine whether the agreement between predicted and true values shown in Figure 5 could have been influenced by climatological patterns encoded in DOY. To assess the relative importance of DOY, together with other input features, we applied the SHAP (SHapley Additive exPlanations) method (Lundberg and Lee, 2017) by computing Shapley values, which represent the average marginal contribution of a feature across all possible feature combinations. This method has been widely used in ML-based atmospheric and remote-sensing studies to evaluate the relative influence of input variables on model predictions (e.g., Lundberg et al., 2020; Zhao et al., 2024). Table 2 summarizes the SHAP values (in %) for the input features—global, direct, and diffuse irradiances, the direct-to-diffuse irradiance ratio, and DOY—for individual absorption bands and for their combined dataset. The SHAP values for the radiative components in Table 2 represent averages over all wavelengths, as reporting wavelength-specific SHAP values would be excessively lengthy. The results clearly show that when global, direct, and diffuse irradiances are used jointly, the global irradiance component consistently exhibits the highest relative SHAP importance (32–48%), followed by the direct and diffuse components. The direct-to-diffuse irradiance ratio contributes a moderate but meaningful amount (approximately 14–22%). In contrast, DOY contributes less than 0.3%, indicating that the seasonal patterns observed in Figure 5 are dominated by spectralirradiance-based features rather than by DOY input feature. DOY can have very low importance because it only provides indirect seasonal information, whereas the spectral irradiances directly capture the actual atmospheric state (water-vapor absorption, scattering, SZA effects, etc.).

**Similarly, we have added the following sentences when discussing the results of Figure 8 in the previous version and Figure 9 in the revised manuscript (611-616).**

Since DOY is included as one of the input features in our DNN model, we also evaluated SHAP values for the input features—global (or direct) irradiance and DOY—when modelling using spectral global or direct irradiances alone, following the same procedure described for Figure 5. In both cases, the irradiance-related feature accounts for more than 97–99% of the total SHAP importance, while the contribution of DOY remains below 2% across all absorption bands. This indicates that the predictive information is overwhelmingly contained in the spectral irradiances themselves, with DOY providing only a very minor contribution.

Since the present work is intended as a feasibility study for DNN-based PWV retrieval, we have not explicitly examined the performance of the DNN model in climatologically distinct regions. However, as explained in our response to Comment 1a, the proposed method can be implemented at other sites through strategies such as retraining or transfer learning when local reference data are available. Future studies aim to evaluate the method across a wider range of atmospheric conditions.

3. What level of uncertainty can be expected or considered acceptable in this context? What are the typical uncertainties associated with the reference instruments, and what level of uncertainty would be tolerable depending on the intended application? Benchmark values should be introduced before discussing the results (e.g., RMSE) and before stating that the model performance is good.

→ The spectral irradiance measurements used for both training and validation were obtained from the same instrument and processed using a consistent procedure. Consequently, any instrument-specific systematic characteristics are implicitly learned by the DNN and largely compensated during model training. Therefore, the dominant source of uncertainty in the predicted PWV is expected to arise from the reference PWV values provided by the MWR, rather than from the irradiance measurements themselves. It is thus reasonable to consider the uncertainty of predicted PWV to be comparable to that of the reference MWR measurements.

Previous studies indicate that integrated water vapor retrievals from ground-based MWRs typically exhibit uncertainties on the order of  $\pm 0.1$ –0.3 cm under favorable conditions, while in less favorable situations uncertainties may be larger (Elgered and Jarlemark, 1998; Minowa et al., 2024; Böck et al., 2025). In this study, the RMSE between observed and predicted PWV values was generally less than 0.24 cm, and in some cases as low as 0.157 cm when all direct, diffuse, and global irradiance components were included (Figure 4). Even when using a single radiation component (global or direct irradiance), RMSE values (Figure 7a) mostly fall below or near the upper bound of  $\pm 0.3$  cm under favorable conditions. These results demonstrate that the proposed method provides reliable PWV estimates under operational conditions, with errors consistent with the expected uncertainty of the reference MWR measurements.

We have summarized the above information in the revised manuscript as given below while discussing the results of the models based on spectral direct, diffuse, and global irradiances (Lines 443-449)

As the spectral irradiance data used for both training and validation were obtained from the same instrument and processed consistently, instrument-specific systematic effects were effectively learned and compensated by the DNN. As a result, the main source of uncertainty in the predicted PWV is likely associated with the reference MWR measurements. Ground-based MWRs typically retrieve PWV with uncertainties of about  $\pm 0.1-0.3$  cm (Elgered and Jarlemark, 1998; Minowa et al., 2024; Böck et al., 2025). The RMSE values obtained in this study were below 0.24 cm, well within this range, indicating that the prediction accuracy is consistent with the expected uncertainty of the standard reference measurements—namely, the MWR observations used in this study.

Similary, we have added below sentences as given below while discussing results corresponding to only global or direct irradiance-based models (Lines 559-561).

These RMSE values are closer to the upper bound of PWV uncertainty corresponding to MWR measurements under favorable conditions (Elgered and Jarlemark, 1998; Minowa et al., 2024; Böck et al., 2025), as discussed in section 4.1.2.1.

- 4. The authors emphasise the need for "continuous retrievals" (line 73). However, the retrieval technique described in the manuscript, which relies on solar measurements, can only be applied during daylight hours, with an unavoidable interruption at night. I could not find any explicit reference to this limitation in the manuscript (although I may have missed it), and it should be clearly acknowledged and discussed.
- → We thank the reviewer for this important comment. In the manuscript, the term "continuous retrievals" is intended to indicate that the proposed method can operate under both clear and cloudy conditions without interruption due to cloud cover, rather than implying 24-hour (day–night) temporal continuity.

To avoid misinterpreation, we have removed the word "continuous" in the revised manuscript. This modification does not affect the interpretation, methodology, or conclusions of the study, but simply clarifies the intended meaning.

- 5. Figures 5 and 8 are useful for illustrating PWV variations across the seasons. However, they are not ideal for assessing the quality of the comparison throughout the year, as the spread of the measurements is large and the whiskers do not provide information about point-to-point correspondence. If the aim is to demonstrate how the agreement between the two instruments varies seasonally, it might be more effective to compute the ratio of each pair of measurements and present a boxplot of that ratio as a function of month.
- → As suggested by the reviewer, we computed box plots for ratio of each pair of measurements as function of month and added in Figure 5 (here Figure R1) and Figure 9 (here Figure R2, which corresponds to Figure 8 of previous version), as shown below.

Figure R1. Monthly comparison of DNN-predicted versus actual PWV values for spectral bands centered at (a) 940 nm, (b) 820 nm, (c) 720 nm, and (d) combination of all of them; and ratios of actual to predicted PWVs for spectral bands centered at (e) 940 nm, (f) 820 nm, (g) 720 nm, and (h) combination of all of them. The box represents the 25th and 75th percentiles, the yellow line inside the box indicates the median, and the whiskers extend to the most extreme points within 1.5 times the interquartile range.

Figure R2. Monthly comparison of predicted and observed PWV values for the spectral band centered at 940 nm, using models based on (a) only global irradiance and (b) only clear-sky direct irradiance as input features; and ratios of actual to predicted values for (d) only global irradiance and (d) only clear-sky direct irradiance as input features. The box represents the 25th and 75th percentiles, the yellow line inside the box indicates the median, and the whiskers extend to the most extreme points within 1.5 times the interquartile range.

We further updated our discussions related to those figures in the revised manuscripts, in sections 4.1.2 and 4.2.2.

Specifically, we added the following sentences in the revised manuscript.

**Lines 413-415**

The corresponding ratio plot (Figure 5e) reinforces this finding. The monthly medians of Actual/Predicted values are nearly constant around 1, and the spreads are narrow, signifying minimal systematic bias and stable retrievals across different atmospheric states.

**Lines 422-424**

The Actual/Predicted ratios (Figure 5f) generally remain close to unity but exhibit slightly broader distributions compared to the 940 nm. Occasional minor departures from 1 indicate small month-specific overestimations or underestimations, likely linked to weaker absorption features at 820 nm

**Lines 431-433**

In the ratio plot (Figure 5g), the median values remain roughly centred around 1, but the interquartile ranges and whiskers are, in general, wider than those in the 940 and 820 nm bands. This broader spread indicates relatively greater variability and lesser stable monthly retrievals than those above two wavelength bands.

**Lines 440-442**

The ratio plot (Figure 5h) shows that the Actual/Predicted medians are tightly centred at around 1 for all months, and the spread remains very narrow, suggesting minimal bias and higher temporal consistency, consistent with the best overall RMSE (0.157 cm) and  $R^2$  (0.982) achieved by this model (Figure 4d).

**Lines 591-593**

The lower panel Figure 9c reinforces this performance. The Actual/Predicted ratios remain centred around 1.0 for all months, with relatively narrow boxes and whiskers, suggesting that the global-irradiance model does not exhibit strong seasonal biases and maintains relatively stable accuracy across the annual cycle.

**Lines 606-609**

This behaviour is also reflected in the ratio plot in Figure 9d. In Figure 9d, although the median ratio remains close to unity, the interquartile ranges and whiskers are, in general, wider compared with those shown in Figure 8c, suggesting relatively larger month-to-month deviations and greater uncertainty in the predictions.

- 6. Are strictly clear-sky conditions required for direct-irradiance retrievals, or is it sufficient that the Sun is not obscured by clouds? Moreover, please explain why clear-sky conditions are necessary for direct irradiance but not for other geometries (line 416), as this may not be immediately evident to all readers.
- → The MS-700 spectroradiometer with a rotating shadowband measures total and diffuse irradiance by cycling through four positions: one with the Sun unblocked to record total irradiance, and three with the Sun blocked at different angles to capture diffuse sky radiation. Direct irradiance is then obtained by subtracting the averaged diffuse measurements from the total (see Equations 1–3). Even if the solar disk is not visibly obscured, clouds near the Sun can scatter additional light into the instrument's field of view, biasing the direct-beam estimate and introducing errors in subsequent PWV retrievals. Therefore, strictly clear-sky conditions are required for direct irradiance-based retrievals. In contrast, while clear-sky conditions can also be used for retrievals from other irradiance components, such as global irradiance, DNNs can learn statistical relationships between spectral global irradiance and PWV under a wider range of atmospheric conditions, including cloudy skies. Consequently, DNN-based retrievals using global irradiance can operate under broader sky conditions, whereas direct-irradiance-based retrievals remain limited to strictly clear-sky scenes. We therefore used direct-irradiance-based retrievals only under clear-sky conditions and other irradiance components for both clear and cloudy skies.

For clarity, we have revised the original sentences as below in the revised manuscript (Lines 506-509).

While this additional screening helped to more strictly ensure clear-sky conditions —necessary because clouds near the solar disk can scatter extra light into the sensor and contaminate the direct-beam measurement, making it difficult to accurately assess the contribution of only direct irradiances in PWV —it also significantly reduced the number of available data samples.

**7. Could the authors more clearly articulate the main advantages of using DNN-based techniques compared with DOAS-type retrievals or radiative transfer calculations?**

→ Thank you for this suggestion. We clarfied the main advantages of using DNN-based techniques compared with DOAS-type retrievals or radiative transfer calculation.

**We have added following sentences in the revised manuscript (Lines 109-115).**

Traditional DOAS-type and fully radiative transfer-based retrievals usually operate within selected wavelength windows, require careful baseline determination, and often assume clear-sky, direct-beam conditions, which limits their applicability under cloudy or diffuse-light scenarios (Irie et al., 2011). In contrast, once a machine-learning model—such as a deep neural network (DNN)—is trained, it can directly map measured spectra to PWV with negligible computational cost, enabling high-temporal-resolution retrievals. By capturing complex, non-linear relationships in the spectral data, ML models also reduce reliance on detailed physical assumptions, offering a flexible, scalable, and robust solution compared with conventional DOAS or radiative-transfer-based methods.

8. A high-temporal-resolution example would be valuable, for instance, a time series of some days showing both the reference dataset and the corresponding DNN retrievals. At present, the paper includes only averaged or summary plots. Including a short time window with pronounced temporal variability (e.g., within a day or over a few days) would help illustrate how smooth or responsive the DNN retrievals are.

→As suggested by the reviewer, we included an example of a high-temporal resultion in the revised manuscript while discussion the comparions between observed and predicited values.

**We have added a figure given below as Figure 6 in the revised manuscript and described it as follow (Lines 453-458).**

Further, to highlight the temporal behaviour of the predictions on shorter time scales, Figure 6 presents example comparisons between predicted and observed PWV for a mostly clear-sky day (April 4, 2018) and a highly cloudy day (June 5, 2018). These examples provide insight into how the models reproduce temporal fluctuations in PWV under contrasting sky conditions. Overall, the predicted and observed values exhibit similar temporal patterns, although small differences in magnitude occasionally appear, particularly under cloudy conditions or in weaker absorption bands. These results further highlight the ability of models to capture PWC variability on shorter time scales.

Figure R3. Temporal variations of DNN-predicted and observed PWV values on April 4, 2018 (mostly clear-sky day) and June 5, 2018 (highly cloudy day).

**TECHNICAL REMARKS**

- Line 22: Please clarify what is meant by "limited data" in this context.

  → We rewrote the sentence to add clarity in the revised mansucript (*Line 23*).
- Lines 29–31: Bibliographic references needed.
   → We added bibliogiraphic references in the revised manuscript (*Line 31*)
- Line 60, "most weather conditions": As this section discusses the limitations of different techniques, it would be important to specify which weather conditions are suitable for MWR measurements.
  - $\rightarrow$  We specified the weather conditions in the revised mansucript (*Line 61-63*)
- Line 171: Correct "cantered" to "centred".
   → We corrected this typo in the revised manuscript (*Line 177*)
- Lines 173–175: At least one relevant bibliographic reference should be added.

  →For clarity, we slight polished the original setnences and then added relevant bibliographic references (*Lines 227-230*)
- Line 207: Please explain why ReLU activation functions were chosen for the DNN architecture.
  - $\rightarrow$  We explianed the reason for using ReLU activation function (*Lines 214-216*).
- Line 232: This is the first occurrence of the term "unseen" in the DNN context, also used in Section 4.1.2. It would be helpful to introduce or define it more clearly.
   →We clarified in the revised manuscript (*Lines 246-247*).
- Line 265: Consider clarifying what is specifically meant by "feature engineering" in this context.
  - $\rightarrow$  We clarified in the revised manuscript (*Lines 295-296*).

- Line 387: Can the "more dynamic conditions" mentioned here be identified or quantified more precisely?
  - → The "more dynamic conditions" may be hard to be quantified using only spectral irradiance data. We, therefore, would like to remove this phrase for clarity and readiblity.
- Line 424: Please specify that all-sky conditions refer to global irradiance, and clear-sky to direct irradiance.
  - → We specified them in the revised mansucript (*Line 177, Table 3*)
- Sections 4.2.2 and 5: The importance of data volume is reiterated several times. Consider removing a few redundant mentions.
  - → We revised those sections and removed redundant mentions.

**References**

- Böck, T., Löffler, M., Marke, T., Pospichal, B., Knist, C., and Löhnert, U.: Instrument uncertainties of network-suitable ground-based microwave radiometers: overview, quantification, and mitigation strategies, Atmospheric Measurement Techniques, 18, 6251–6270, 2025.
- Chen, D., Guo, H., Gu, X., Wang, J., Liu, Y., Li, Y. & Wu, Y.: Physical-guided transfer deep neural network for high-resolution AOD retrieval, Remote Sens., 17, 3606, 2025.
- De Mazière, M., Thompson, A. M., Kurylo, M. J., Wild, J. D., Bernath, P. F., Blumenstock, T., Hannigan, J. W., Lambert, A., Mahieu, E., and McGee, T. J.: The Network for the Detection of Atmospheric Composition Change (NDACC): history, status, and perspectives, Atmos. Chem. Phys., 18, 4935–4964, 2018.
- Dong, S., Li, Y., Zhang, Z., Gou, T. & Xie, M.: A transfer-learning-based windspeed estimation on the ocean surface: Implication for the spatial-spectral resolution of remote sensors, Appl. Intell., 54, 7603–7620, 2024.
- Elgered, G. and Jarlemark, P. O. J.: Ground-based microwave radiometry and long-term observations of atmospheric water vapor, Radio Science, 33, 707–717, 1998.
- Gupta, S., et al.: Spatial transfer learning for estimating PM2.5 in data-sparse regions, Proc. ACM, 2024.
- Irie, H., Takashima, H., Kanaya, Y., Boersma, K. F., Gast, L., Wittrock, F., Brunner, D., Zhou, Y., and Van Roozendael, M.: Eight-component retrievals from ground-based MAX-DOAS observations, Atmos. Meas. Tech., 4, 1027–1044, 2011.
- Lundberg, S. M. and Lee, S.-I.: A unified approach to interpreting model predictions, Adv. Neural Inf. Process. Syst., 30, 2017.
- Lundberg, S. M., Erion, G., Chen, H., DeGrave, A., Prutkin, J. M., Nair, B., Katz, R., Himmelfarb, J., Bansal, N., and Lee, S.-I.: From local explanations to global understanding with explainable AI for trees, Nat. Mach. Intell., 2, 56–67, 2020.
- Minowa, M., Araki, K., and Takashima, Y.: Compact microwave radiometer for water vapor estimation with machine learning method, SOLA, 20, 339–346, 2024.
- Mizobuchi, S., Irie, H., and Shimizu, S.: Long-term continuous observations of the horizontal inhomogeneity in lower-atmospheric water vapor concentration using A-SKY/MAX-DOAS, Prog. Earth Planet. Sci., 12, 24, 2025.
- Pan, S. J. and Yang, Q.: A survey on transfer learning, *IEEE Trans. Knowl. Data Eng.*, 22, 1345–1359,2010.
- Platt, U. and Stutz, J.: Differential Optical Absorption Spectroscopy: Principles and Applications, Springer, Berlin, 2008.
- Weiss, K., Khoshgoftaar, T. M., and Wang, D.: A survey of transfer learning, *J. Big Data*, 3, 9, 2016.
- Zhao, Y., Zhou, Y., Xia, H., Cui, C., Chen, Z., and Huang, J.: Studying the aerosol effect on deep convective clouds over the global oceans by applying machine learning techniques on long-term satellite observation, Remote Sens., 16, 2487, 2024.

---

## Author Comment (AC2)

**Replies to comments of Reviewer 2**

Authors would like to express sincere gratitude to an anonymous reviewer for his/her valuable comments and suggestions. Our manuscript has greatly benefited from these insights. We have carefully revised the manuscript, taking all comments into account. Our responses to the reviewer's comments are provided below.

**General comment**

This study provides practical insights for estimating PWV from the spectral measurements of solar global, direct, and diffuse irradiances. The retrieval of PWV under all-sky conditions is useful for monitoring the atmospheric conditions. However, concerns exist regarding the following points: The input data for the DNN model includes "Day number of year" and "solar zenith angle". The model may have learned seasonal characteristics of the observation site. If the instrument is relocated to another site, would retraining be necessary?

→ Thank you for this valuable comment. In our approach, solar zenith angle (SZA) is not used as an independent input feature. Instead, all spectral irradiances are first normalized by the cosine of SZA, so that the geometric effect of solar incidence is already incorporated into the transformed radiometric features before they are provided to the DNN. Thus, the model does not receive SZA as a separate predictor; its influence is implicitly captured through the SZA-normalized spectra.

To assess whether the model nevertheless relies on seasonal or site-specific characteristics—particularly through the inclusion of DOY—we evaluated the contribution of all input variables using the SHAP framework (Lundberg and Lee, 2017). SHAP provides a unified, theoretically grounded framework based on cooperative game theory to quantify the contribution of each input feature to the model output by computing Shapley values, defined as the average marginal contribution of a feature across all possible feature combinations. The method has been widely used in recent ML-based atmospheric and remote-sensing studies to assess the relative influence of input variables on model predictions (e.g., Lundberg et al., 2020; Zhao et al., 2024). Tables R1-R3 below summarize the SHAP values obtained for all input features under different modeling strategies (i.e., using global, direct, and diffuse irradiances jointly, or using only global or only direct irradiances), as described in the manuscript. The SHAP values for the radiative components shown in the table represent averages across all wavelengths, given that wavelength-by-wavelength SHAP reporting would be excessively lengthy.

Table R1. Relative SHAP values (in %) of the input features for the DNN models trained using spectral direct, diffuse, and global irradiances of different absorption bands, together with the direct-to-diffuse irradiance ratio (Ratio) and the day of year (DOY)

| Centered band/Features | Direct(%) | Diffuse(%) | Global(%) | Ratio(%) | DOY(%) |  |
|------------------------|-----------|------------|-----------|----------|--------|--|
| 940 nm                 | 23.8      | 17.89      | 35.67     | 22.42    | 0.22   |  |
| 820 nm                 | 18.3      | 19.2       | 48.39     | 13.97    | 0.14   |  |
| 720 nm                 | 20.03     | 15.24      | 44.69     | 19.95    | 0.08   |  |
| Combined (all above)   | 27.18     | 19.43      | 32.92     | 20.35    | 0.11   |  |

Table R2. Relative SHAP values (in %) of the input features for the DNN models trained using spectral global irradiances of different absorption bands together with day of year (DOY)

| Centered band/Features | Global(%) | DOY(%) |
|------------------------|-----------|--------|
| 940 nm                 | 98.65     | 1.35   |
| 820 nm                 | 99.67     | 0.33   |
| 720 nm                 | 99.51     | 0.49   |
| Combined (all above)   | 99.54     | 0.46   |

Table R3. Relative SHAP values (in %) of the input features for the DNN models trained using spectral direct irradiances of different absorption bands together with day of year (DOY)

| Centered band/Features | Global(%) | DOY(%) |
|------------------------|-----------|--------|
| 940 nm                 | 97.84     | 2.16   |
| 820 nm                 | 99.19     | 0.81   |
| 720 nm                 | 99.15     | 0.85   |
| Combined (all above)   | 98.49     | 1.51   |

Across all model configurations (Tables R1–R3), spectral irradiances account for more than 95% of the total feature importance. DOY contributes less than 2% (often below 0.3%), indicating that the model's predictive power is drawn almost entirely from the radiometric information. Because the irradiances are normalized by SZA, and because spectral absorption and scattering signatures inherently vary with atmospheric path length, the effects of SZA and seasonal variability are already encoded in the spectral measurements themselves. This explains the negligible contribution of DOY and shows that the model is not learning seasonal characteristics of the training site.

We agree that the model does not explicitly distinguish site-dependent and instrument-dependent factors; it learns statistical relationships between irradiances and PWV. As a result, relocating the same instrument to a site with markedly different atmospheric or surface conditions may reduce retrieval accuracy if the model is applied without adaptation. Nevertheless, this does not limit the broader applicability of the approach, as the model can be effectively extended to new locations either by retraining with locally available PWV references (e.g., radiosonde or GNSS PWV) or by applying transfer learning, in which a pre-trained model is fine-tuned using a relatively small amount of local irradiance–PWV data (Pan et al., 2010; Weiss et al., 2016). Recent studies (e.g., Chen et al., 2025; Dong et al., 2024; Gupta et al., 2024) demonstrate that such adaptation strategies successfully maintain retrieval performance when transferring ML-based remotesensing models across sites.

We have summarized above discussion in the revised manuscirpt by including Table R1 in the revised mansucript as Table 2, and then adding following sentences (Lines 459-474).

Since DOY is included as one of the input features in our DNN models, it is important to examine whether the agreement between predicted and true values shown in Figure 5 could have been

influenced by climatological patterns encoded in DOY. To assess the relative importance of DOY, together with other input features, we applied the SHAP (SHapley Additive exPlanations) method (Lundberg and Lee, 2017) by computing Shapley values, which represent the average marginal contribution of a feature across all possible feature combinations. This method has been widely used in ML-based atmospheric and remote-sensing studies to evaluate the relative influence of input variables on model predictions (e.g., Lundberg et al., 2020; Zhao et al., 2024). Table 2 summarizes the SHAP values (in %) for the input features—global, direct, and diffuse irradiances, the direct-to-diffuse irradiance ratio, and DOY—for individual absorption bands and for their combined dataset. The SHAP values for the radiative components in Table 2 represent averages over all wavelengths, as reporting wavelength-specific SHAP values would be excessively lengthy. The results clearly show that when global, direct, and diffuse irradiances are used jointly, the global irradiance component consistently exhibits the highest relative SHAP importance (32–48%), followed by the direct and diffuse components. The direct-to-diffuse irradiance ratio contributes a moderate but meaningful amount (approximately 14–22%). In contrast, DOY contributes less than 0.3%, indicating that the seasonal patterns observed in Figure 5 are dominated by spectralirradiance-based features rather than by DOY input feature. DOY can have very low importance because it only provides indirect seasonal information, whereas the spectral irradiances directly capture the actual atmospheric state (water-vapor absorption, scattering, SZA effects, etc.).

**Again, we have added the following sentences to clarify the query related to the application over a new environment (Lines 264-270).**

Furthermore, the model learns statistical relationships between spectral irradiances and PWV, which can be influenced by local atmospheric and surface conditions. Relocating the instrument to a site with substantially different conditions may reduce accuracy. However, the model can be adapted to new locations through retraining with local reference PWV data or, more efficiently, via transfer learning (Pan et al., 2010; Weiss et al., 2016), in which a pre-trained model is fine-tuned using a relatively small amount of site-specific data. Recent studies (e.g., Chen et al., 2025; Dong et al., 2024; Gupta et al., 2024) show that such adaptation strategies effectively preserve retrieval performance when transferring ML-based remote-sensing models across sites.

The methodology details and results are clearly described. This paper is recommended after minor revision.

**Specific comments**

- •L50-53: GNSS precise point positioning is also used to retrieve PWV in worldwide (e.g., Zhang et al., 2019, DOI:10.1109/JSTARS.2019.2906950).
- → Thank you for providing this valuable information.

**We have updated our manuscript by adding the following sentences in the revised mauscript (Lines 72-75).**

In parallel, the International GNSS Service (IGS) Real-Time Pilot Project has enhanced global access to high-precision GNSS data streams, enabling near-real-time estimation of PWV through precise point positioning (PPP) techniques at dense ground-based networks worldwide (Zhang et

al., 2019).

- Figure 2: How did you obtain the calibration constants at each wavelength? Is the radiometric calibration necessary in the retrieval method of this study?
- $\rightarrow$  The calibration constants at each wavelength were obtained through the factory calibration conducted by EKO Instruments using a NIST-traceable standard lamp. These calibration coefficients are embedded in the instrument's calibration file and are automatically applied by the operating software to convert raw detector outputs into physical spectral irradiance values (Wm-2µm-1). Therefore, since the measurements are already provided in calibrated physical units, no additional radiometric calibration is required in the retrieval method used in this study.

**We have added the following sentence in the revised maucript to add clarity (Lines 145-147)**

These spectral irradiances, in units of  $Wm^{-2}\mu m^{-1}$ , are automatically generated by the observational software from the raw signals after applying the factory calibration constants.

• L220-229: There are many technical terms. If possible, include the references.
→ Thank you for this important suggestion.

We have added relevant references in the revised manuscript (Lines 229-245).

- L493-494: Could the discrepancies in Fig. 8b from July to September be attributed to the limited amount of training data available during the wet season?
- → Yes, the discrepancies observed in Figure 8b (Figure 9b in the revised manuscript) from July to September can indeed be attributed to the limited amount of training data available during the wet season. During this period, the frequency of cloud cover is substantially higher (Khatri and Takamura, 2009), which significantly reduces the availability of direct solar irradiance measurements required for training. Consequently, the model may have encountered fewer representative samples under clear-sky conditions, leading to reduced predictive accuracy and greater variability in the retrieval results for those months.

**We have added the following sentence in the revised maucript to add clarity (Lines 604-606)**

In particular, the discrepancies in Figure 9b from July to September likely stem from the limited training data during the wet season, when persistent cloud cover greatly reduces the number of usable direct irradiance measurements (Khatri and Takamura, 2009).

**References**

- Chen, D., Guo, H., Gu, X., Wang, J., Liu, Y., Li, Y. & Wu, Y.: Physical-guided transfer deep neural network for high-resolution AOD retrieval, Remote Sens., 17, 3606, 2025.
- Dong, S., Li, Y., Zhang, Z., Gou, T. & Xie, M.: A transfer-learning-based windspeed estimation on the ocean surface: Implication for the spatial-spectral resolution of remote sensors, Appl. Intell., 54, 7603–7620, 2024.
- Gupta, S., et al.: Spatial transfer learning for estimating PM2.5 in data-sparse regions, Proc. ACM, 2024.
- Lundberg, S. M. and Lee, S.-I.: A unified approach to interpreting model predictions, Adv. Neural Inf. Process. Syst., 30, 2017.
- Lundberg, S. M., Erion, G., Chen, H., DeGrave, A., Prutkin, J. M., Nair, B., Katz, R., Himmelfarb, J., Bansal, N., and Lee, S.-I.: From local explanations to global understanding with explainable AI for trees, Nat. Mach. Intell., 2, 56–67, 2020.
- Khatri, P. & Takamura, T.: An algorithm to screen cloud-affected data for sky radiometer data analysis, J. Meteorol. Soc. Japan, 87, 189–204, 2009.
- Pan, S. J. and Yang, Q.: A survey on transfer learning, *IEEE Trans. Knowl. Data Eng.*, 22, 1345–1359,2010.
- Weiss, K., Khoshgoftaar, T. M., and Wang, D.: A survey of transfer learning, *J. Big Data*, 3, 9, 2016.
- Zhang, H., Yuan, Y., Li, W., and Zhang, B.: A real-time precipitable water vapor monitoring system using the national GNSS network of China: method and preliminary results, *IEEE J. Sel. Top. Appl. Earth Obs. Remote Sens.*, **12**, 1587–1598,2019.